# The Role of the FGF19 Family in the Pathogenesis of Gestational Diabetes: A Narrative Review

**DOI:** 10.3390/ijms242417298

**Published:** 2023-12-09

**Authors:** Agata Sadowska, Elżbieta Poniedziałek-Czajkowska, Radzisław Mierzyński

**Affiliations:** Chair and Department of Obstetrics and Perinatology, Medical University of Lublin, Jaczewskiego 8, 20-090 Lublin, Poland; agata.sadowska@yahoo.com (A.S.); radekm1969@gmail.com (R.M.)

**Keywords:** fibroblast growth factors, FGF19 family, pregnancy, gestational diabetes, diabetes mellitus, metabolic disorders

## Abstract

Gestational diabetes mellitus (GDM) is one of the most common pregnancy complications. Understanding the pathogenesis and appropriate diagnosis of GDM enables the implementation of early interventions during pregnancy that reduce the risk of maternal and fetal complications. At the same time, it provides opportunities to prevent diabetes, metabolic syndrome, and cardiovascular diseases in women with GDM and their offspring in the future. Fibroblast growth factors (FGFs) represent a heterogeneous family of signaling proteins which play a vital role in cell proliferation and differentiation, repair of damaged tissues, wound healing, angiogenesis, and mitogenesis and also affect the regulation of carbohydrate, lipid, and hormone metabolism. Abnormalities in the signaling function of FGFs may lead to numerous pathological conditions, including metabolic diseases. The FGF19 subfamily, also known as atypical FGFs, which includes FGF19, FGF21, and FGF23, is essential in regulating metabolic homeostasis and acts as a hormone while entering the systemic circulation. Many studies have pointed to the involvement of the FGF19 subfamily in the pathogenesis of metabolic diseases, including GDM, although the results are inconclusive. FGF19 and FGF21 are thought to be associated with insulin resistance, an essential element in the pathogenesis of GDM. FGF21 may influence placental metabolism and thus contribute to fetal growth and metabolism regulation. The observed relationship between FGF21 and increased birth weight could suggest a potential role for FGF21 in predicting future metabolic abnormalities in children born to women with GDM. In this group of patients, different mechanisms may contribute to an increased risk of cardiovascular diseases in women in later life, and FGF23 appears to be their promising early predictor. This study aims to present a comprehensive review of the FGF19 subfamily, emphasizing its role in GDM and predicting its long-term metabolic consequences for mothers and their offspring.

## 1. Introduction

Gestational diabetes mellitus (GDM) is defined as a carbohydrate metabolic disorder of any degree of severity that is first diagnosed during pregnancy [1]. It is one of the most common complications during pregnancy. The prevalence of GDM is reported in a wide range, from 1 to 14% in North American countries, with the most common being in the 3–8% range [2,3,4]. The overall weighted prevalence of GDM in 24 European countries was estimated at 10.9% [5]. The ADA (American Diabetes Association) recognizes that GDM occurs in 7% of pregnant women in the United States [6]. The global prevalence of GDM is estimated at 3–6% [7], although according to the latest data from the IDF (International Diabetes Federation), it affects about 14% of pregnancies worldwide resulting in the number of approximately 18 million pregnant women per year [8]. Typically, GDM, which ends with the delivery according to criteria set by the WHO (World Health Organization), is most often diagnosed at 24–28 weeks of pregnancy after an oral 75 g glucose load test [9]. Gestational diabetes usually affects women who have risk factors such as GDM in previous pregnancies, chronic hypertension, age > 35 years, overweight or obesity, having a baby weighing more than 4 kg, having more than two children, having a newborn with a malformation, a history of intrauterine deaths, a family history of type 2 diabetes, and polycystic ovary syndrome [8].

Abnormalities of carbohydrate tolerance during pregnancy significantly impact the course of pregnancy and can lead to a broad spectrum of maternal and fetal complications [10]. Available evidence shows that in utero hyperglycemia plays a significant role in the subsequent development of the child, increasing the risk of obesity, metabolic syndrome, and type 2 diabetes [11]. Understanding the pathogenesis and correctly diagnosing GDM allows treatment implementation to prevent several maternal and fetal complications. It also creates opportunities to avoid diabetes, metabolic syndrome, and cardiovascular disease in women and their offspring in later life [12]. It is essential during the current epidemic of carbohydrate disorders and obesity in women. Recent years have brought much information about the genetic and molecular determinants of GDM. Genomewide association studies (GWAS) were described to detect associations between single nucleotide polymorphisms (SNPs) and diseases with complex pathophysiological backgrounds, including GDM [13]. The study by Kwak et al. showed that genetic variants in CDKAL1 and near MTNR1B are strongly associated with GDM. The same loci are known to be associated with T2DM, which confirms the similar genetic basis of T2DM (Type 2 diabetes mellitus) and GDM [14,15].

Several reports highlight the possibility of using fibroblast growth factors (FGFs) as potential early biomarkers in diagnosing metabolic diseases. However, studies on the role of FGFs in the pathogenesis of GDM are scarce, and their results are inconclusive. FGFs represent a heterogeneous family of signaling proteins that can stimulate fibroblast growth. FGFs play a vital role in cell proliferation and differentiation, repair of damaged tissues, wound healing, angiogenesis, and mitogenesis [16]. FGFs also affect the regulation of carbohydrate, water–electrolyte, lipid, and hormone metabolism [17]. Abnormalities in the signaling function of FGFs may lead to numerous pathological conditions, such as cancer and metabolic diseases [18].

This review aims to provide insight into the current state of knowledge on the importance of FGFs and their role in the pathogenesis of gestational diabetes.

## 2. Scope and Methodology

This review aims to present the current state of knowledge on the mechanisms of action of the FGF19 family and its involvement in the pathogenesis of gestational diabetes. The list of relevant bibliographic references was determined based on PubMed and MEDLINE databases, using MeSH terms as follows: ‘fibroblast growth factors’, ‘FGF19 family’, ‘FGF19’, ‘FGF 21’, ‘FGF 23’, ‘pregnancy’, ‘gestational diabetes’, ‘diabetes mellitus’, and ‘metabolic disorders’ as keywords. Databases were searched until August 2023. Only articles in English were considered.

## 3. Fibroblast Growth Factors

Fibroblast growth factors are produced in many organs and tissues, such as the small intestine, colon, liver, adipose tissue, brain, kidney, and parathyroid glands [19,20]. The multiplicity of responses elicited and the tissues in which they are synthesized account for FGFs’ great possible diagnostic and therapeutic potential. The human genome encodes 22 proteins belonging to the FGF family, which is divided into seven subfamilies based on homologous sequence and mechanism of action [16,20]. All proteins in this family contain a conserved core domain composed of 120 amino acids [18]. Most FGFs bind to and activate fibroblast growth factor receptors (FGFRs) on the cell surface, showing a high affinity for heparan sulfate proteoglycans (HSPG), which act as cofactors [19]. The signal transduction process is accomplished by binding to a transmembrane FGFR, which is formed of four parts: three extracellular immunoglobulin-like domains (D1–D3) and an intracellular domain that acts as a tyrosine kinase [16]. These types of FGFs are called canonical and have autocrine and paracrine effects [19].

The FGF19 subfamily, also known as atypical FGFs, which includes FGF19, FGF21, and FGF23, is essential in regulating metabolic homeostasis [21,22]. Factors belonging to this group have limited affinity for heparan sulfate (HS) found in the extracellular space. It is due to changes within the proteoglycan-binding domain, resulting in a lack of hydrogen bond formation between heparan sulfate and the amino acids of the binding domain [23]. Thus, the FGF19 family factors can enter the systemic circulation and act as hormones [23,24].

To initiate the endocrine pathway, members of the FGF19 subfamily require coreceptors, proteins belonging to the Klotho group [25,26,27,28]. Klotho is a family of single-pass transmembrane glycoproteins, which includes α-Klotho, β-Klotho, and γ-Klotho, referred to as KLPH (KL lactase phlorizin hydrolase) or LCTL (lactase-like protein) [19,29]. The *Klotho (KL)* gene was first identified in 1997 in transgenic mice [30]. The gene and protein are named after the Greek goddess Klotho, who, along with her sisters, spun the thread of human life and determined its length [31]. The human *KL* gene is located on chromosome 13q12 and has five exons. The Klotho protein exists in two forms—transmembrane and secretory—and each has distinct functions [32]. Klotho isoforms are expressed in various tissues and cell types and affect the activity of metabolic processes within them [33]. The secretory form of the protein is involved in inhibiting the insulin/IGF-1 (insulin-like growth factor 1) pathway. Mice deficient in Klotho protein show hypoglycemia and high insulin sensitivity, while moderate insulin and IGF-1 resistance was observed in *Klotho* overexpression [34].

### 3.1. FGF19

FGF19 was first identified in 1999 in the human brain during fetal development [35]. There are no precise data in the literature describing the structure of FGF19. Harmer et al. presented the structural biology of FGF19 [36]. The lack of arrangement in the structure of the HS-binding segment between the β10 and β12 strands led to the suggestion that the topology of the HS19 binding site is inherently flexible [36]. According to another conception, both copies of the factor exhibited a well-ordered β10–β12 region [24]. The β1-β2 loop of FGF19 is the longest among the FGFs, and unlike paracrine ligands, there is a spatial separation between the β1–β2 loop and the β10–β12 segment [37].

FGF19 is expressed mainly in the small intestine and colon enterocytes and has insulin-like effects [18,20]. Sites of its expression include the skin, gallbladder, kidney, and umbilical cord [38]. Izzaguirre et al. also showed that FGF19, in combination with the KLB (β Klotho) complex, activated an insulin-independent endocrine pathway in metabolic organs such as the liver, adipose tissue, and brain [19]. The C-terminal domain of FGF19 is an essential region for Klotho protein recognition, while the N-terminal region is vital for interactions with FGFR [39].

Schmidt et al. found that regulating FGF19 synthesis was mediated by cholesterol and fat-soluble vitamins A and D [11]. Diet1 protein was identified as the first to transcriptionally and post-transcriptionally modulate FGF19 expression levels. *Diet1* encodes a 236 kD protein and is expressed in small intestinal enterocytes. It influences the production of FGF19 in humans through the regulation of CYP7A1 (cholesterol 7 α—hydroxylase) [40].

#### Mechanisms of Action

It is believed that FGF19 can increase hepatic glycogen synthesis in an insulin-independent manner by inducing phosphorylation and inactivation of GSK3 (glycogen synthase kinase-3) [41]. FGF19 also influences the homeostasis maintenance of carbohydrate metabolism by inhibiting gluconeogenesis through repression of CREB (cAMP-response element-binding protein), a key regulator of gluconeogenic genes such as PGC1alpha (peroxisome proliferator-activated receptor gamma coactivator 1-α) [42].

The superior centers involved in regulating hunger and satiety are located in the brain. The primary I-level center, the arcuate nucleus, is in the hypothalamus. It consists of two populations of oppositely acting neurons: AgRP/NPY (Agouti-related protein/neuropeptide Y), which stimulate food intake, and POMC/CART (pro-opiomelanocortin/amphetamine-regulated transcript), which are responsible for satiety and appetite inhibition [43]. A study published by Morton et al. showed that FGF19 crossed the blood–brain barrier and, after binding to FGFR1, reduced AgRP/NPY activity. It is thought that FGF19 might directly improve glucose metabolism through this mechanism [44,45,46]. Ryan et al. observed that intraventricular infusion of FGF19 was associated with reduced food intake and improved glucose metabolism in lean and diet-induced obese rats [46].

FGF19 is a negative regulator of bile acid feedback [47]. It has a regulatory function on various cellular effects in response to an ingested meal [48]. FGF19 expression is induced by activation of FXR (farnesoid X receptor), which regulates fatty acid biosynthesis and circulation [21,49,50]. In response to bile acid absorption, the intestine produces FGF19, inhibiting CYP7A1, the first enzyme in the bile acid biosynthetic pathway [51,52]. In the liver, FGF19 inhibits bile acid synthesis and gluconeogenesis and, by inhibiting CYP7A1, also affects hepatocytes directly through the FGFR4 and β-Klotho receptor complex. Thus, FGF19 influences the synthesis rate of bile acids and is indispensable in the postprandial release of bile from the gallbladder [53,54,55,56].

FGF19 increases energy expenditure and insulin sensitivity in adipose tissue by binding to FGFR1 and FGFR4, influencing body weight and glucose concentrations [19,57].

The metabolic effects of FGF19 are summarized in Figure 1.

### 3.2. FGF21

Fibroblast growth factor 21 is considered a multifunctional cytokine released in response to high levels of carbohydrates. It is crucial in regulating glucose and lipid metabolism and maintaining the body’s energy homeostasis [58]. Some studies have shown that circulating FGF21 in humans exhibits a characteristic diurnal rhythm, with a significant increase occurring between midnight and early morning [59]. FGF21 consists of 210 amino acids; its gene is located on chromosome 19 in the 5′ region. The DNA sequence of human FGF21 gene is 75% similar to mouse DNA and 35% similar to that of FGF19 [60,61]. The results of recent studies suggested that FGF21 may be an effective drug for treating metabolic disorders [62,63]. FGF21 is an adipokine produced mainly by liver cells, adipose tissue, and pancreas [64,65], but it is also expressed in skeletal muscle, heart, kidney, testes, and pancreas [60]. It is a factor detected in blood serum, confirming its endocrine function [66]. FGF21 binds to the β-Klotho protein in complex with FGFR1, 2, and 3 and activates the Ras proteins/mitogen-activated protein kinases (Ras/MAPK) signaling cascade [53,67,68].

The study published by Lee et al. reported that other nuclear receptors, such as the H-binding protein CREB, could also regulate FGF21 expression [69]. Boutant et al. suggested a link between FGF21 and SIRT1 (silent information regulator 1), which is thought to result from PPARα (peroxisomes proliferators-activated receptor α) induction [70]. SIRT1 is an NAD+ (nicotinamide adenine dinucleotide)-dependent deacetylase that regulates gene expression through histone deacetylation [71]. It is considered a potent regulator of cellular metabolism by preventing oxidative stress and inflammation [71,72]. There is evidence that SIRT1 may enhance the insulin sensitivity of tissues such as the liver, skeletal muscle, and adipose tissue and may induce insulin secretion [73,74].

#### Mechanisms of Action

The studies by Kharitonenkov et al. revealed the ability of FGF21 to improve glycemic control through insulin-independent glucose uptake by adipocytes, which is caused by increased expression of GLUT1 (glucose transporter protein type 1) [21,75]. This results in a decrease in blood glucose and triglyceride levels [21]. In adipose tissue, FGF21 expression is upregulated via the peroxisomes proliferators-activated receptor γ (PPARγ) under various physiological conditions, including fasting and feeding a high-fat diet [65]. PPARα stimulates liver FGF21 expression under fasting conditions [76,77,78]. Pathophysiological factors, such as ketogenic diet, overfeeding, cold, hepatic oxidative stress, and mitochondrial stress in skeletal muscle, are important in regulating FGF21 synthesis [76,78,79,80,81]. Recent studies also revealed the stimulatory effect of 4ATF4 (activating transcription factor 4) on FGF21 expression and glucocorticoid receptors [82]. In humans, metabolic upregulation of the PPARα receptor has been shown to lead to increased circulating FGF21 levels [83,84]. FGF21 improves pancreatic β-cell function and viability by activating p44/42 protein kinase [81]. Chen and colleagues found that FGF21 not only increased insulin secretion and insulin content in pancreatic islets but also protected pancreatic β-cells from apoptosis through activation of ERK ½ (extracellular signal-regulated kinase) and Akt (protein kinase B) signaling pathways [85].

Incretins such as GLP-1 (glucagon-like peptide-1) and GIP (glucose-dependent insulinotropic polypeptide, previously called gastric inhibitory polypeptide) are the endocrine system parts involved in postprandial stimulation of insulin secretion and physiological regulation of glucose homeostasis [86,87].

GLP-1 analogues enhance the hepatic synthesis of FGF21, while blocking FGF21 attenuates the inhibitory effect of GLP-1 analogues on hepatic glucose output [88,89]. The study by Pan et al. investigated the synergistic actions of FGF21 with the incretin glucagon-like peptide-1 (GLP-1) by generating GLP-1-Fc-FGF21 dual agonists. One of the dual agonists, GLP-1-Fc-FGF21 D1, exerted a robust and long-lasting glucose-lowering effect in diabetic mice. In addition, it improved body weight and lipid profile to a greater extent than using GLP-1 or FGF21 alone [90]. FGF21 levels correlate with various cardiometabolic conditions, and higher FGF21 expression is associated with cardiovascular risk factors such as chronic inflammation, obesity, hypertriglyceridemia, and elevated liver enzymes [91,92]. Studies on the effects of FGF21 on myocardial cells are inconclusive. The investigations have shown an association between elevated FGF21 levels and hypertension, cardiac disease, myocardial infarction, atrial fibrillation, and cardiac atrial fibrosis [93,94,95].

Elevated FGF21 levels have also been demonstrated among patients with type II diabetes, and it is believed that this factor may be associated with a significant increase in the risk of set-associated complications in this group of patients [96,97]. In contrast, other studies have shown that FGF21 secreted in an autocrine mechanism protects against oxidative stress and prevents hypertrophy and the effects of coronary remodeling [98,99,100,101]. So far, data on the association of polymorphisms in the FGF21 signaling pathway with GDM are lacking. However, it has been found that common genetic variations in FGFR2 may be associated with metabolic risk expressed by LDL cholesterol [102].

The cardioprotective properties of FGF21 in myocardial infarction, ischemia-reperfusion injury, or intense beta-adrenergic stimulation have been found in animal models and myocardial cell cultures [98,103]. Other animal experimental findings have also shown that hyperglycemia and reduced FGF21 levels could enhance myocardial damage, inflammation, and apoptosis [101,104]. Tanajak et al. proved that long-term administration of FGF21 to rats with insulin resistance and obesity markedly improved tissue insulin sensitivity and myocardial function through the activation of antiapoptotic and mitochondrial signaling pathways [101].

The metabolic effects of FGF21 are presented in Figure 2.

### 3.3. FGF23

FGF23 is a 32 kDa polypeptide with an N-terminal and C-terminal region consisting of 251 amino acids, synthesized mainly by osteocytes and osteoblasts in response to high concentrations of dietary phosphate intake or associated with high vitamin D concentrations [26,105,106]. The target organs of this cytokine are the kidneys and parathyroid glands [27]. FGF23 expression has also been detected in the salivary glands, stomach, and tissues such as skeletal muscle, brain, mammary gland, liver, lung, spleen, and heart [20].

#### Mechanisms of Action

Recent studies demonstrated a role of FGF23 in processes regulating the body’s energy homeostasis [107,108]. FGF23 is associated with inflammation through IL-6 (interleukin 6), IL-10 (interleukin 10), and CRP (C-reactive protein) and with symptoms of metabolic syndrome: insulin resistance, visceral obesity, and dyslipidemia [107,109,110]. Insulin is a hormone that, in addition to affecting carbohydrate metabolism, is also involved in phosphate homeostasis. Because of the overlapping sites of action of insulin and FGF23 in the kidneys, attempts have been made to evaluate the association of FGF23 with insulin resistance in patients with advanced stages of chronic kidney disease. The study has found significantly higher levels of ctFGF23 (carboxyl-terminal FGF23) in patients with concomitant insulin resistance [111]. Similarly, Wahl et al. reported that diabetes was significantly associated with increased FGF23 levels in patients with chronic kidney disease [112]. Hanks et al. showed a positive correlation between FGF23 levels and HOMA-IR (homeostatic model assessment for insulin resistance); the positive association also persisted in patients without chronic kidney disease [107]. Since the conclusions remain discrepant, additional studies are needed to investigate the association of FGF23 with insulin resistance and its effects on carbohydrate metabolism [109,113].

FGF23 regulates serum phosphate and vitamin D concentrations through the FGFR-α-Klotho complex, which is highly expressed in the kidney [27,114,115]. The endocrine effect of FGF23 on phosphate involves the downregulation of NaPi-2a (apical type IIa sodium-dependent phosphorus cotransporter) and NaPi-2c (apical type IIc sodium-dependent phosphorus cotransporter) in the proximal tubule of the nephron, leading to increased phosphate excretion and parathormone inhibition [116,117,118]. Furthermore, FGF23 decreases the activity of α-hydroxylase involved in calcitriol synthesis. It increases the activity of the competitive enzyme 24-α-hydroxylase, which synthesizes the inactive form of vitamin D [116]. Thus, FGF21 could be found as a hormone involved in calcium-phosphate homeostasis.

Data reported that increased levels of FGF23 in patients with chronic kidney disease may be responsible for left ventricular hypertrophy [119]. FGF23 requires the cofactor Klotho to activate the signaling pathway [114]. Soluble Klotho has been shown to protect the heart by inhibiting TRPC6 (transient receptor potential cation channel 6), the overexpression of which leads to cardiac hypertrophy and remodeling [120]. It has been revealed that FGF23-mediated decreased vitamin D concentrations may enhance angiotensin II secretion and affect sodium metabolism [121,122,123,124,125]. It indicates that FGF23 may indirectly affect the development of hypertension. It was confirmed by Andrukhova et al., who showed that FGF23 increased renal sodium reabsorption and thus may contribute to hypertension and cardiac hypertrophy development [124].

The metabolic effects of FGF23 are presented in Figure 3.

## 4. FGF19 Family in GDM

With the increasing number of pregnant women with GDM, the attention of researchers is focused on understanding the exact mechanisms of the development of this pregnancy complication, which would allow the introduction of early effective identification of women at risk of developing GDM and the implementation of preventive interventions, which has significant clinical relevance. Given the promising results of studies evaluating the involvement of the FGF19 family in the pathogenesis of carbohydrate metabolism disorders, their role in GDM is being investigated. However, studies conducted to date have not yielded unequivocal results.

In a study published by Wang et al., the mRNA and protein expression levels of FGF19 and FGF21 in patients with GDM and healthy ones were examined. β Klotho receptor was also determined in the placenta, rectus abdominis muscles, and subcutaneous adipose tissue. It was found that women with GDM had significantly lower expressions of mRNA and FGF19 protein in the placenta and rectus muscles compared to the control group, while the expression of FGF21 and KLB showed no significant differences. In subcutaneous adipose tissue, the expression of FGF19 and FGF21 was undetectable. The authors also noted the higher birth weight and placental weight in the GDM group [125]. Colomiere et al. suggested that this may result from a postreceptor defect in signaling pathways for insulin in the placentas of women with GDM and obesity [126]. FGF19’s inhibitory effect on bile acid synthesis and gluconeogenesis in the liver has been noticed. Schaefer-Graf et al. confirmed that free fatty acids strongly correlated with fetal growth in pregnant patients with GDM [127].

Wang et al. found a close negative correlation between FGF19 expression and a positive correlation of FGF21 expression with insulin resistance. They suggested that reduced FGF19 expression in women with GDM may be related to the pathophysiology of the disease, while elevated FGF21 expression could result from a compensatory response to the insulin resistance observed in GDM. They also demonstrated decreased FGF19 levels during the second trimester in patients with GDM compared to controls [18]. Administration of human recombinant FGF19 to pregnant high-fat-diet mice was observed to reduce fasting blood glucose, HOMA-IR, triglycerides, and free fatty acids. In addition, upregulation of the expression of p-IRS1 (placental-insulin receptor substrate 1) and GLUT4 was found in the placentas of the experimental animals. These findings could suggest that the administration of FGF19 might improve glycolipid metabolism and alleviate systemic and local insulin resistance. It has been considered that the upregulation of placental expression of p-IRS1 and GLUT4 may be responsible for the beneficial metabolic effect of FGF19 [128].

Yang et al. published a report on the relationship between FGF19 and GDM, fetal growth (birth weight and neonatal length), and sex. It was the first study to evaluate the concentration of FGF19 in cord blood. Concentrations of insulin, proinsulin, C-peptide, insulin-like growth factor I (IGF-I), and insulin-like growth factor II (IGF-II) were also evaluated. No differences in the concentrations in cord blood plasma of FGF19, proinsulin, and C-peptide between pregnant women with GDM and healthy ones were found. FGF19 levels were similar in male and female newborns and were higher in newborns born prematurely compared to those born at term. FGF19 levels were higher in newborns born by cesarean section than those born by vaginal delivery. FGF19 was positively correlated with neonatal size only in female newborns, suggesting a sex-specific effect of FGF19 on fetal growth [129]. The study by Karasek et al. showed that when gestational diabetes is diagnosed early (8–14 weeks), FGF19 concentrations do not differ significantly between diabetic and healthy pregnant women throughout pregnancy. The authors showed a negative correlation between FGF19 concentration body, weight, and BMI (Body Mass Index). They believe that in the case of an early diagnosis of GDM, including diet and lifestyle changes may have prevented a decrease in FGF19 concentrations in the second and third trimesters, which is expected to explain the lack of differences in FGF19 concentrations between groups. However, the study was conducted in tiny groups [130].

In the report by Wang et al., FGF19 and FGF21 levels in mothers with GDM were significantly associated with insulin resistance and polycystic ovary syndrome. Observation of FGF19 and FGF21 levels during the second trimester of pregnancy showed reduced FGF19 and elevated FGF21 levels in GDM patients. In contrast, serum concentrations of adiponectin, which is involved in carbohydrate metabolism and fatty acids in the liver and muscle, were significantly lower in GDM patients [18].

The results of this study are consistent with the observations of Stein et al., who found a positive correlation between FGF21 levels and fasting insulin concentrations, HOMA-IR, and triglyceride levels. Negative correlations were shown between FGF21 levels and adiponectin and HDL (high-density lipoprotein) levels. However, patients with GDM did not have higher serum FGF21 concentrations than healthy ones [131]. The report published by Li et al. revealed significantly higher FGF21 levels in white blood cells and serum in the GDM group compared to the control one at 28 weeks’ gestation [132]. These observations were confirmed by Jia et al., who showed significantly higher FGF21 levels in GDM patients than in healthy controls. In addition, they observed that FGF21 levels in this group were related to prepregnancy BMI, weight gain during pregnancy, leptin, RBP-4 (retinol binding protein 4), and adiponectin in GDM, although leptin, RBP-4, and adiponectin levels were comparable in both groups [133].

The study by Bonakdaran et al. evaluating FGF21 also showed significantly higher levels in the GDM group than in the control group. A cut-off of 82.07 ng/L of FGF21 had a sensitivity of 100% and specificity of 85% for the prediction of GDM. The authors did not find an association between FGF21 levels and insulin resistance. These observations supported the significance of FGF21 in the development of GDM [134].

A recently published meta-analysis of nine papers (827 subjects, 397 cases of GDM, and 435 controls) confirmed the association of higher FGF21 concentrations with GDM (random effects (MD), 95%CI = 80.46, [0.07–0.86], *p* = 0.02) [135].

In the nested case-control study by Wang et al., serum FGF21 levels were determined at 14–21 weeks of gestation, and FGF21 concentrations were significantly higher in the group diagnosed with GDM later in pregnancy than in the normal glucose tolerance group. Additionally, the association of FGF21 and GDM was assessed using logistic regression analysis by stratifying study subjects into quartiles for FGF21 levels (Q1 ≤ 27.83 pg/mL, Q2 28.00 to 45.82 pg/mL, Q3 46.81 to 83.87 pg/mL, Q4 ≥ 84.09 pg/mL). FGF21 quartiles Q2, Q3, and Q4 were associated with greater odds of GDM occurrence than Q1 after multivariable adjustments. The authors of this investigation concluded that measurement of FGF21 in the second trimester may be a potentially helpful new biomarker for the early identification of pregnant women at risk of developing GDM [136]. Similar conclusions were reached by Wu et al., who, in a large prospective study, showed that elevated FGF21 concentrations in early pregnancy (6–15 weeks) were associated with an increased risk of developing GDM, especially in overweight and obese women [137].

The first study to evaluate FGF21 concentrations in human umbilical cord blood and its effect on postnatal fetal growth in pregnancy complicated by GDM was published by Megia et al. The children’s weight, height, and body mass index Z-score (BMI ZS) were assessed at birth and 12, 24, and 48 months. Maternal FGF21 concentrations (mFGF21) were higher in patients with GDM than in healthy pregnant women, while cord blood FGF21 concentrations (cbFGF21) were similar in both groups. It was also shown that mFGF21 concentrations were significantly higher than cbFGF21 levels. In the GDM group, mFGF21 levels were positively correlated with the HOMA-IR index and insulin and glucose levels during the 75 g oral glucose load test. In addition, a positive correlation was observed between prepregnancy BMI and maternal triglyceride levels. The above study showed that cbFGF21 levels in both groups were positively correlated with mFGF21 levels, while a negative correlation was found between cbFGF21 levels and total cholesterol and HDL levels. The authors of this study also revealed that cbFGF21 concentrations significantly correlated with neonatal BMI in both groups at 12 and 24 months of age [138].

Other results were presented by Xu et al., who showed that the gene and protein expression of FGF21 in adipose tissue cells was reduced, and FGF21 serum levels were decreased in women with GDM compared to the healthy ones. The authors of this study believe that decreased FGF21 levels are associated with the risk of GDM, which may suggest its potential significance as a diagnostic marker for the development of GDM [139].

The study by Mosavat et al. aimed to assess the association of serum FGF21 and FGF23 with the risk of GDM. It was revealed that FGF21 and FGF23 concentrations were lower in GDM patients compared to healthy ones. FGF23 levels were inversely associated with GDM, so a 1 μg/mL decrease in FGF23 levels was associated with a 1.4-fold increased risk of developing GDM. There was no association of FGF21 concentrations with the risk of GDM development. The authors believe that lower FGF23 concentrations could be involved in the pathophysiology of GDM, but FGF21, even though associated with metabolic risk factors in pregnancy, may not be a fundamental factor in GDM [140]. Distinct findings were presented by Kizilgul et al., who found that women with GDM had significantly higher serum FGF23 levels than the control group. Serum FGF23 level was positively correlated with BMI, insulin concentrations, and HOMA-IR [141].

Hocher et al. evaluated the concentration of the intact FGF23 molecule (iFGF23) and the cumulative concentration of the intact FGF23 and degradation products of FGF23 (cFGF23) in random mother/child pairs. Pregnant women with GDM had higher iFGF23 concentrations than healthy pregnant women, while cFGF23 concentrations were not significantly different. Multivariate regression analyses showed that GDM was associated with increased iFGF23 concentrations irrespective of confounding factors such as age, BMI, ethnic background, family history of diabetes, smoking during pregnancy, and recurrent pregnancy loss. The study failed to show a difference in iFGF23 and cFGF23 concentrations in the cord blood of newborns of GDM mothers and non-GDM patients [142].

These observations were confirmed by Tuzun et al., who showed significantly higher levels of FGF23 in GDM. In this study, carotid intima-media thickness (cIMT), left ventricle (LV) mass, LV mass index, and myocardial perfusion imaging (MPI) as markers of atherosclerosis and cardiac dysfunction in GDM patients were also evaluated. All these parameters were significantly higher in the GDM group, and FGF23 was positively correlated with CIMT, LV mass index, and MPI. It might suggest that monitoring serum FGF23 levels may be helpful as an early, noninvasive indicator of subclinical atherosclerosis in patients with GDM [143].

Dekker et al. evaluated the placental expression of FGF21, its coreceptors, and PPARα and PPARγ in normoglycemic pregnancies and pregnancies complicated by GDM. The study also examined FGF21 levels in maternal serum, umbilical cord blood, and GLUT-1, -3, and -4 expressions. The target sites of FGF21 expression in the placenta were syncytiotrophoblast, endothelial, stroma, and Hofbauer cells. Women with GDM had a significantly higher median expression of FGF21 in the placenta than women in the control group. FGF21 mRNA expression was not associated with maternal BMI before pregnancy. There were no significant differences in the expression of receptor isoforms for FGF21, while β Klotho, GLUT3, and GLUT4 showed increased expression in GDM. PPARα expression was higher in women with GDM, while there was no difference in PPARγ expression. No significant variations in maternal serum FGF21 levels in GDM patients compared to normoglycemic pregnant women were found. However, the expression of FGF21 protein alone was higher in the study group compared to the control. FGF21 was undetected in cord blood from patients in either group. The expression of FGF21 and its receptors in the placenta suggests that it may be an essential factor influencing placental metabolism, thereby contributing to fetal growth and metabolism regulation in pregnancy complicated by GDM [144]. 

The results of studies on the association of FGFs with GDM are presented in Table 1.

Gestational diabetes is significantly associated with an increased risk of developing type 2 diabetes mellitus (T2DM) and future cardiovascular complications in both mother and fetus. According to preliminary observations, FGF23 appears to be a promising noninvasive marker for developing vascular complications following GDM [141,143]. Whether elevations in FGF21 persist after delivery in GDM women at increased risk for long-term metabolic and vascular dysfunction has not been well studied. The study by Durnwald et al. conducted in women with metabolic disorders (T2DM, metabolic syndrome) and healthy women who had pregnancies complicated by GDM in the previous 5–10 years showed no differences in FGF21 levels [145].

## 5. Conclusions

Many mechanisms and factors responsible for developing gestational diabetes have been proposed. Fibroblast growth factors are a family of signaling proteins whose impairment can lead to several conditions, including disorders of carbohydrate metabolism. The research results on the importance of FGFs in GDM are sparse, primarily pointing to their involvement in the pathogenesis of this pregnancy complication and their potential use as early predictive factors. However, they are inconclusive, and many issues, such as the type of samples and processing methods, have still to be elucidated and standardized.

FGF19 and FGF21 have been shown to be associated with insulin resistance, which suggests their potential relevance as predictive factors for GDM development. The expression of FGF21 and its receptors in the placenta may imply their impact on placental metabolism and thus contribute to regulating fetal growth and metabolism. The observed relationship between FGF21 and increased birth weight could be suggestive of a potential role for FGF21 in predicting future metabolic abnormalities in children born to women with GDM. In this group of patients, different mechanisms may contribute to an increased risk of cardiovascular disease in later life, and FGF23 appears to be a promising early predictor of these complications.

Despite the promising results of studies on the role of FGFs in the pathogenesis of GDM and predicting its late consequences in mother and child, little is known about the factors that regulate its functions and activity. Therefore, further research on FGFs’ molecular mechanisms of action and prognostic significance seems warranted.

## Figures and Tables

**Figure 1 ijms-24-17298-f001:**
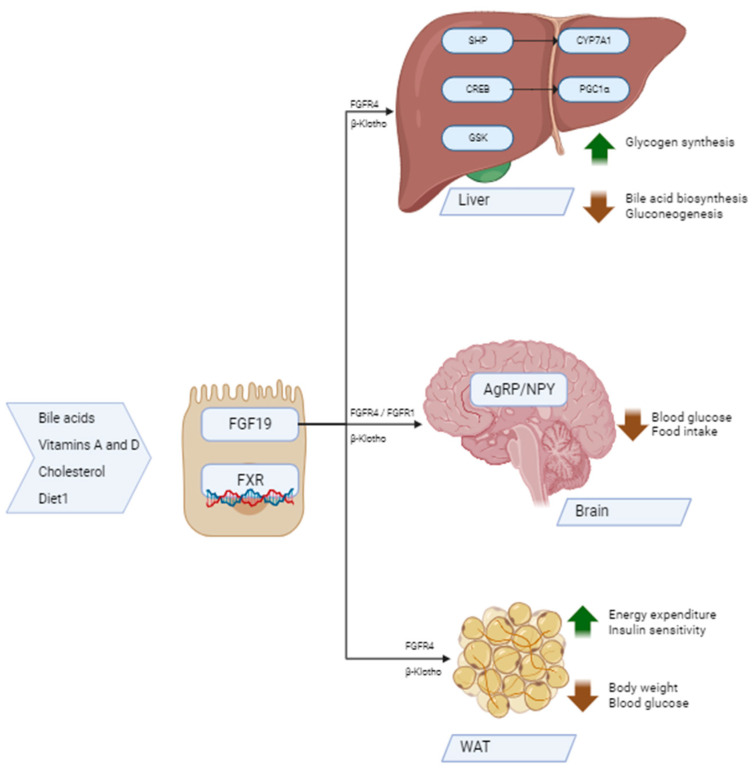
Metabolic effects of FGF19. Abbreviations: FGF19—fibroblast growth factor 19; FXR—farnesoid X receptor; FGFR4—fibroblast growth factor receptor 4; FGFR1—fibroblast growth factor receptor 1; β-Klotho—Klotho β-like protein; SHP—nuclear small heterodimer partner; CYP7A1—cholesterol 7 α hydroxylase; CREB—cAMP-response element-binding protein; PGC1α—peroxisome proliferator-activated receptor γ coactivator 1-α; GSK—glycogen synthase kinase; ↑—increased; ↓—decreased; AgRP—Agouti-related protein; NPY—neuropeptide Y; WAT—white adipose tissue.

**Figure 2 ijms-24-17298-f002:**
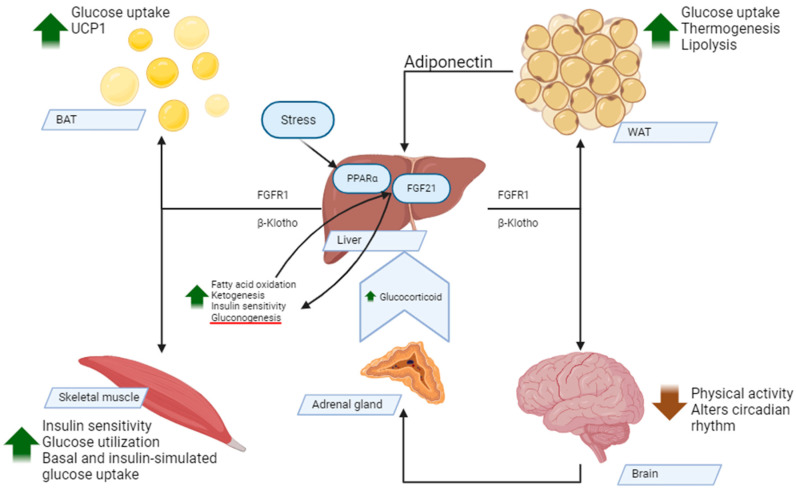
Metabolic effects of FGF21. Abbreviations: FGF21—fibroblast growth factor 21; PPARα—peroxisomes proliferators-activated receptor α; FGFR1—fibroblast growth factor receptor 1; β Klotho—Klotho β-like protein; WAT—white adipose tissue; BAT—brown adipose tissue; UCP1—uncoupling protein 1.

**Figure 3 ijms-24-17298-f003:**
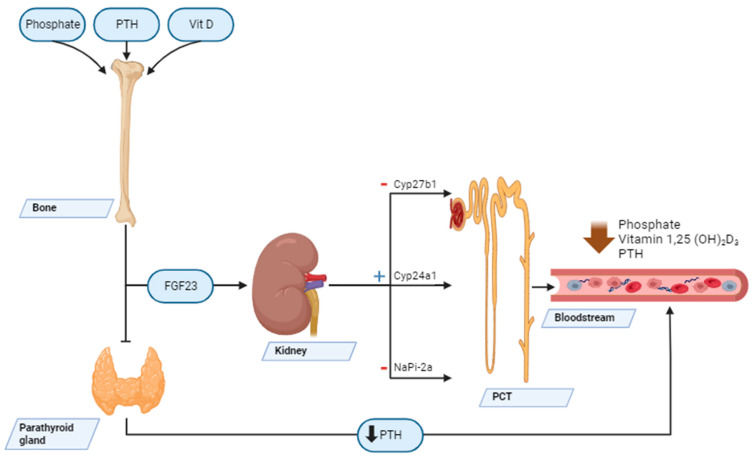
Metabolic effects of FGF23. Abbreviations: FGF23—fibroblast growth factor; PTH—parathyroid hormone; PCT—proximal convoluted tubules; Cyp27b1—25-hydroxyvitamin D-1α-hydroxylase; Cyp24a1—24α-hydroxylase; NaPi-2a—sodium-dependent phosphate cotransporter 2A.

**Table 1 ijms-24-17298-t001:** The association of FGFs with GDM.

Study	Size of the Groups	Effects		
		FGF19	FGF21	FGF23
Wang et al. [125]	20 GDM 25 healthy	Significantly lower expression in the placenta and rectus muscle in GDM.	No significant differences in subcutaneous adipose tissue.	X
Wang et al. [18]	30 GDM 60 healthy	Close negative correlation with insulin resistance. Reduced levels in GDM (2nd trimester).	Positive correlation with insulin resistance. Elevated levels in GDM (2nd trimester).	X
Yang et al. [129]	153 GDM 153 healthy	Higher concentration in cord blood in newborns born prematurely and by elective cesarean section.	X	X
Karasek et al. [130]	23 GDM 54 healthy	Concentration levels in early diagnosed GDM do not differ significantly from healthy patients. Negative correlation with BMI and body weight.	X	X
Stein et al. [131]	40 GDM 80 healthy	X	Positive correlation with fasting insulin levels, HOMA-IR, triglyceride levels. Negative correlation with adiponectin and HDL. No difference between GDM and healthy patients.	X
Li et al. [132]	51 GDM 50 healthy	X	Higher levels in white blood cells and serum in GDM.	X
Jia et al. [133]	62 GDM 58 healthy	X	Significantly higher levels of FGF21 concentrations in GDM. Relation to prepregnancy BMI in GDM.	X
Bonakdaran et al. [134]	30 GDM 60 healthy	X	Significantly higher levels of FGF21 concentrations in GDM. No correlation with insulin resistance.	X
Jia et al. [135]	397 GDM 435 healthy	X	Positive correlation of higher FGF21 concentrations with GDM.	X
Wang et al. [136]	133 GDM 133 healthy	X	FGF21 quartiles Q2, Q3, and Q4 were associated with greater odds of GDM occurrence than Q1.	X
Wu et al. [137]	332 GDM 664 healthy	X	Elevated FGF21 concentrations in early pregnancy (6–15 weeks) were associated with an increased risk of developing GDM, especially in overweight and obese women.	X
Megia et al. [138]	79 GDM 78 healthy	X	Maternal concentration higher in GDM. mFGF21 significantly higher than cbFGF21 in GDM. mFGF21 levels positively correlated with HOMA-IR and insulin and glucose levels. Positive correlation between cbFGF21 and BMI of children at 12 and 24 months.	X
Kizilgul et al. [141]	46 GDM 36 healthy	X	X	Higher levels in GDM.
Mosavat et al. [140]	53 GDM 43 healthy	X	Significantly lower concentration in GDM (24–28 weeks). No differences before and after delivery.	Lower concentration in GDM (24–28 weeks) and before delivery. 1 pg/mL decrease in concentration associated with a 1.4× increased risk of GDM.
Hocher et al. [142]	64 GDM 762 healthy	X	X	No difference in iFGF23 and cFGF23 concentrations in the cord blood of newborns of GDM mothers and non-GDM patients.
Tuzun et al. [143]	54 GDM 33 healthy	X	X	Significantly higher concentrations in GDM.

Abbreviations: GDM—gestational diabetes mellitus; FGF19—fibroblast growth factor 19; FGF21—fibroblast growth factor 21; FGF23—fibroblast growth factor 23; HOMA-IR—homeostatic model assessment for insulin resistance; HDL—high-density lipoprotein; mFGF21—maternal fibroblast growth factor 21; cbFGF21—cord blood fibroblast growth factor 21; BMI—body mass index, X—not studied.

## Data Availability

No new data were created or analyzed in this study. Data sharing is not applicable to this article.

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
