# Peer review of "The Role of the FGF19 Family in the Pathogenesis of Gestational Diabetes: A Narrative Review"

_ijms, 2023, doi:10.3390/ijms242417298_

Round 1
Reviewer 1 Report
Comments and Suggestions for Authors
The authors provide a narrative review on the relationship between FGF19 family members and GDM pathophysiology.
The overall rationale is clear.
Introduction: Fine.
Methods: The search algorithm is too specific; wildcards and searching for FGF19, FGF21 and FGF23 in addition to the other search terms is recommended. Presenting the actual full search algorithm would be warranted.
Main text: Will be evaluated after confirmation, that the accessible literature has already been found with the existing search algorithm.
Comments on the Quality of English Language
minor changes needed
Author Response
We would like to thank you very much for your commitment and contribution to reviewing our work. We are very grateful for all the comments and suggestions that will significantly improve the quality of our manuscript.
The authors provide a narrative review on the relationship between FGF19 family members and GDM pathophysiology.
The overall rationale is clear.
Introduction: Fine.
Methods: The search algorithm is too specific; wildcards and searching for FGF19, FGF21 and FGF23 in addition to the other search terms is recommended. Presenting the actual full search algorithm would be warranted.
We have supplemented the subsection" Scope and methodology" with the following keywords: FGF19, FGF21, FGF23. In the search for relevant articles for our manuscript, these were used separately with the keyword gestational diabetes. Again, the available literature was reviewed with the above algorithm applied, and the manuscript was supplemented with the papers found.
Main text: Will be evaluated after confirmation, that the accessible literature has already been found with the existing search algorithm.
Reviewer 2 Report
Comments and Suggestions for Authors
In the present study, the authors conducted an extensive literature review on the role of FGF19 family in pathogenesis of gestational diabetes.
The manuscript addresses a relevant topic, and it is also clear and well written.
One point that I would like to see addressed in genetic susceptibility for gestational diabetes. Are there genome wide association studies (GWAS) showing the association between specific SNPs (or variants) on FGF with gestational diabetes?
Author Response
We would like to thank you very much for your commitment and contribution to reviewing our work. We are very grateful for all the comments and suggestions that will significantly improve the quality of our manuscript.
In the present study, the authors conducted an extensive literature review on the role of FGF19 family in pathogenesis of gestational diabetes.
The manuscript addresses a relevant topic, and it is also clear and well written.
One point that I would like to see addressed in genetic susceptibility for gestational diabetes. Are there genome wide association studies (GWAS) showing the association between specific SNPs (or variants) on FGF with gestational diabetes?
We have addressed the above issue with reference to the literature.
Reviewer 3 Report
Comments and Suggestions for Authors
The authors focused on the pathophysiological roles of fibroblast growth factor: FGF subfamily in the development of gestational diabetes mellitus: GDM and make a summary of the clinical significance as follows:
1. FGF19 and FGF21 are associated with insulin resistance, which suggests their potential relevance as predictive factors for GDM development.
2. FGF21 and its receptors are responsible for the metabolism of the placenta, fetal growth, and metabolism.
3. Children born to women with GDM can be predicted to have future metabolic abnormalities through FGF21.
4. For individuals with GDM who are at risk of developing cardiovascular disease later in life, FGF23 has the capability of early prediction of these complications.
The topic discussed in this review article is crucial in understanding the pathophysiology of GDM, but previous studies have found significant associations between FGF subfamily and GLP-1, so the authors need to address these issues.
Author Response
We would like to thank you very much for your commitment and contribution to reviewing our work. We are very grateful for all the comments and suggestions that will significantly improve the quality of our manuscript.
The authors focused on the pathophysiological roles of fibroblast growth factor: FGF subfamily in the development of gestational diabetes mellitus: GDM and make a summary of the clinical significance as follows:
- FGF19 and FGF21 are associated with insulin resistance, which suggests their potential relevance as predictive factors for GDM development.
- FGF21 and its receptors are responsible for the metabolism of the placenta, fetal growth, and metabolism.
- Children born to women with GDM can be predicted to have future metabolic abnormalities through FGF21.
- For individuals with GDM who are at risk of developing cardiovascular disease later in life, FGF23 has the capability of early prediction of these complications.
The topic discussed in this review article is crucial in understanding the pathophysiology of GDM, but previous studies have found significant associations between FGF subfamily and GLP-1, so the authors need to address these issues.
We have addressed the above issue with reference to the literature.
Round 2
Reviewer 1 Report
Comments and Suggestions for Authors
The authors have revised their manuscript in accordance to the reviewer's suggestions. I consider it ready for acceptance.
Reviewer 3 Report
Comments and Suggestions for Authors
I have no further comment.